# Economic Evaluation of the Thai Diagnostic Autism Scale for Autism Spectrum Disorder Diagnosis in Children Aged 1–5 Years Old

**DOI:** 10.3390/healthcare12070782

**Published:** 2024-04-03

**Authors:** Duangkamol Tangviriyapaiboon, Unchalee Permsuwan, Chosita Pavasuthipaisit, Athithan Sriminipun, Piyameth Dilokthornsakul

**Affiliations:** 1Rajanagarindra Institute of Child Development, Chiang Mai 50180, Thailand; yingricd@gmail.com (D.T.); drchosita@gmail.com (C.P.); athithan.srim@cmu.ac.th (A.S.); 2Center for Medical and Health Technology Assessment (CM-HTA), Department of Pharmaceutical Care, Faculty of Pharmacy, Chiang Mai University, Chiang Mai 50200, Thailand; unchalee.permsuwan@cmu.ac.th

**Keywords:** Thai diagnostic autism scale, autism spectrum disorder, economic evaluation, children

## Abstract

The Thai Diagnostic Autism Scale (TDAS) was developed for autism spectrum disorder (ASD) diagnosis in Thai children aged 1–5 years. Previous studies have indicated its good performance; however, additional health resources and healthcare providers are necessary for evaluation. Therefore, this study aimed to assess the cost-effectiveness of TDAS compared to clinical diagnosis (ClinDx) for ASD diagnosis in Thai children aged 1–5 years from a societal perspective. The analysis employed a hybrid model consisting of a decision tree model for a diagnostic phase with a state transition model for a follow-up phase. A literature review was conducted to determine TDAS performance and the relative risk of death in patients with ASD. Direct medical costs were assessed through a retrospective medical records review, and a cross-sectional survey was conducted to determine direct nonmedical costs, ASD severities, and utility values. The cost of TDAS was derived from a healthcare provider interview (n = 10). The incremental cost-effectiveness ratio (ICER) compared the total lifetime cost and quality-adjusted life years (QALY) between TDAS and ClinDx. We found that TDAS could improve QALY by 1.96 but increased total lifetime cost by 5577 USD, resulting in an ICER of 2852 USD/QALY. Sensitivity analysis indicated an 81.16% chance that TDAS is cost-effective. The probabilities of different ASD severities were key influencing factors of the findings. In conclusion, TDAS is the cost-effective option for ASD diagnosis in Thai children aged 1–5 years compared to ClinDx, despite some uncertainties around inputs. Further monitoring and evaluation are warranted if TDAS is to be implemented nationwide.

## 1. Introduction

Autism Spectrum Disorder (ASD) is a complex neurodevelopmental condition characterized by persistent deficits in social communication and interaction, along with restricted and repetitive patterns of behavior, interests, or activities [1]. This multifaceted disorder varies significantly among individuals in terms of symptoms and severity levels.

ASD typically manifests in early childhood, often before the age of three, and significantly impacts the individual’s ability to engage in reciprocal social relationships and communicate effectively [1,2,3].

The etiology of ASD is multifactorial, involving genetic, environmental, and neurological factors. Various genetic markers are associated with ASD, emphasizing the heritability of the disorder [4,5]. Environmental factors, such as prenatal complications, exposure to certain toxins, and perinatal complications, have also been implicated in ASD development [6].

A recent systematic review indicates that the median global prevalence of ASD is 100 patients per 10,000 population, ranging from 1.09 to 436.0 patients per 10,000 population. Males are approximately 4.2 times more likely to be affected than females. The review also reveals the higher ASD prevalence in Africa and the Americas compared to Western Pacific and Southeast Asia [7]. In Thailand, there have been various sources of data for the prevalence of ASD. According to the updated country profiles of autism in ASEAN from the Department of Empowerment of Persons with Disabilities indicates that 0.60% of patients with a disability registered in the database have autism [8].

One of the key factors for successful ASD treatment is early intervention, which can improve children’s overall function [9]. Therefore, the early detection of ASD in high-risk children is crucial to enabling timely access to early intervention. Various diagnostic tools have been employed for ASD detection, such as the Autism Diagnostic Observation Schedule, second edition (ADOS-2) [10], the Childhood Autism Rating Scale, second edition (CARS^TM^2) [11], and the Autism Diagnostic Interview-Revised (ADI-R) [12]. Previous cost-effectiveness studies have shown that early detection and providing interventions before definite ASD diagnosis could enhance long-term clinical outcomes and be cost-effective, especially for a high-risk population [13,14,15].

In Thailand, the ADOS-2 is one of the ASD diagnosis tools to identify children with ASD. However, there are concerns regarding the validity of the translated version of the Western diagnostic tools due to cultural differences, and its use is limited to particular settings. According to the limitations of ADOS-2, clinical diagnosis (ClinDx) based on the fifth edition of the Diagnostic and Statistical Manual of Mental Disorders (DSM-5) remains the standard practice for ASD diagnosis in the country [16].

A novel ASD diagnostic tool, the Thai Diagnostic Autism Scale (TDAS), has been developed and validated for assessing children suspected of having ASD [17,18]. TDAS comprises 13 observation items and 17 interview items. TDAS is a valid tool to diagnose children with ASD with a sensitivity of 82.86% and a specificity of 80.93%. Despite its accuracy of 82.05%, TDAS requires additional resources, such as healthcare providers, instruments during observation, and cloud storage for recording video during observation, resulting in an additional healthcare cost for ASD detection. To date, information on the cost-effectiveness of TDAS for ASD diagnosis is not available. Cost-effectiveness information is important information for national healthcare policy decision-making. At the time of conducting this study, the National Health Security Office (NHSO), a public healthcare payer in the Thai healthcare system covering approximately 70% of Thai citizens, is considering including TDAS into its health benefit packages for Thai citizens under the Universal Coverage Schemes (UCS). The NHSO requires economic evidence, especially cost-effectiveness analysis, to evaluate the value for money of TDAS for ASD diagnosis. This study aimed to assess the cost-effectiveness of TDAS compared to ClinDx for ASD diagnosis for Thai children aged 1–5 years from a societal perspective using a standard decision tree with a Markov model.

## 2. Materials and Methods

### 2.1. Overall Description

A cost-utility analysis was undertaken from a societal perspective with a lifetime horizon, utilizing a hybrid model that combined a decision tree model and a cohort-based state transition model to represent the clinical pathway of suspected ASD patients aged 1–5 years. The suspected patients were children identified with development delays in receptive language (RL), expressive language (EL), and personal and social skills (PS) from the Thai Early Developmental Assessment for Intervention (TEDA4I) [19] or the Developmental Surveillance and Promotion Manual (DSPM) [20] or patients with positive ASD screening by the Pervasive Development Disorders Screening Questionnaires (PDDSQ) [21,22]. The model was built using Microsoft Excel^®^ 365, adhering to the Thai Health Technology Assessment Guideline 2019 and the Consolidated Health Economic Evaluation Reporting Standards 2022 [23,24]. Informed consent was obtained from all the subjects prior to participation. The study aligns with the Declaration of Helsinki, and the protocol was approved by the Ethics Committee of the Department of Mental Health, Ministry of Public Health (No. DMH.IRB COA 005/2566).

### 2.2. Intervention and Comparator

The intervention of interest was TDAS, which consisted of two sections. The first section consisted of 13 items for behavioral observation, recording children’s behaviors in communication, social interaction, play, and repetitive behaviors. The second section involved interviewing parents or caregivers about the children’s development and behaviors. TDAS required administration by trained healthcare providers. According to previous studies [17,18], the sensitivity and specificity of TDAS were 82.86% and 80.93%, respectively, and the accuracy was 82.05%. The comparator was ClinDx of ASD in children, which was based on the fifth edition of the Diagnostic and Statistical Manual of Mental Disorders (DSM-5) [25,26].

### 2.3. Model Structure and Assumptions

A two-phase model was applied: a diagnostic phase and a follow-up phase (Figure 1). During the diagnostic phase, a decision tree model was employed. Either TDAS or ClinDx was used to diagnose suspected patients. Patients with TDAS could be identified as test-positive and test-negative, with TDAS-positive patients further distinguished as true positives and false positives. True positive patients were categorized by ASD severity as mild, moderate, or severe, whereas false positive patients were considered non-ASD patients. Conversely, patients diagnosed with ClinDx were classified as ASD (having the disease) and non-ASD. ASD patients were further classified into clinically confirmed or delayed diagnosis. Patients with clinically confirmed ASD could be classified as mild, moderate, or severe, while patients with delayed diagnosis were assumed to have severe ASD. The severity of ASD was defined according to DSM-5 criteria: level 1 (requiring minimal support) for mild ASD, level 2 (requiring some support) for moderate ASD, and level 3 (requiring most support for daily living) for severe ASD. The delayed diagnosis was defined as a new ASD diagnosis after the age of five.

During the follow-up phase, a Markov model was implemented. Patients entered the Markov model as mild, moderate, or severe ASD according to their severity in the diagnostic phase. They could transition between these health states or to an absorbing state (death), with a one-year cycle length.

### 2.4. Model Validation

This model was clinically validated through the first stakeholder meeting prior to the study’s initiation. The scope and proposed model structure of the study were presented to the stakeholders, whose suggestions were then collected, summarized, and applied to the model as appropriate. Stakeholders included three pediatricians specializing in child developmental and behavior, three child and adolescent psychiatrists, a general physician, three health economists, a representative of parents of children with ASD, and representatives from Thai health system payers. The final model structure received approval from a child’s developmental pediatrician and a child and adolescent psychiatrist. Additionally, the model codes were reviewed and verified by two health economists to ensure the model’s coding accuracy.

### 2.5. Model Inputs

#### 2.5.1. The Performance of TDAS and Probabilities of Being Diagnosed

A pragmatic review literature was conducted to determine the performance of TDAS. The performance of TDAS was derived from a previous study [17]. The probability of a true positive of TDAS was 82.86%, whereas the probability of a false negative was 19.61%. The likelihood of being diagnosed by TDAS was 89.90% [27], compared to 67.31% for ClinDx [17] (Table 1).

#### 2.5.2. Transitional Probabilities and Relative Risk of Death in Patients with ASD

A retrospective medical records review was conducted across five hospitals where TDAS was piloted, to determine the probabilities of being classified as mild, moderate, or severe ASD at the time of ASD diagnosis by either TDAS or ClinDx. The hospitals included one specialty hospital for child development, one tertiary hospital, one large general hospital, one medium-size general hospital, and one district hospital. Transitional probabilities of the changes across severity levels were calculated by survival analyses. A total of 295 patients were included in the medical records review, with 147 patients diagnosed using TDAS and 148 patients diagnosed using ClinDx. Of these, 121 patients (41.02%) were from the specialty hospital for child development, 17 (5.76%) and 51 patients (17.29%) were from the tertiary hospital and the district hospital, and the remainder were from the two general hospitals. The average age was 3.52 ± 1.84, with 81.3% being males. The probabilities of being classified as having mild, moderate, and severe ASD by TDAS were 11.1%, 50.7%, and 38.2, respectively, while those by ClinDx were 6.1%, 33.1%, and 60.8%, respectively. The transitional probabilities from mild to moderate ASD were 6.4% per year, and those from moderate to severe ASD were 2.2%. The details of transitional probabilities are presented in Table 1. However, the relative risk of death in patients with ASD was taken from previous literature [28].

#### 2.5.3. Costs and Health Utility

A retrospective database analysis was performed to determine the direct medical cost associated with ASD treatment. The analysis utilized administrative databases from the five hospitals across the country abovementioned, incorporating inpatient and outpatient medical charges. These databases were integrated with the baseline characteristics and severity data from a retrospective medical records review, aiding in the determination of transitional probabilities across different severities and changing across health states.

Concurrently, a cross-sectional interview was conducted to evaluate the direct non-medical cost and utility values for patients, involving the same individuals whose medical records were reviewed. In total, 295 caregivers of ASD patients were interviewed by trained study coordinators using structured interview forms to gather cost data. The EQ-5D-Y was employed to assess patient utilities. Due to the unavailability of EQ-5D-Y crosswalk for Thai children, a Japanese crosswalk was applied instead, reflecting a cultural comparable setting [29]. In addition, interviews with healthcare providers were also carried out to assess the labor cost associated with diagnosing ASD using TDAS and ClinDx, along with the cost of TDAS implementation and maintenance.

Briefly, the outpatient direct medical cost of ASD treatment was 66 US dollars (USD) per year for mild ASD patients, while it was 113 and 124 USD per year for moderate and severe ASD patients. Annual inpatient direct medical costs were 387, 416, and 664 USD for mild, moderate, and severe ASD patients. The cost of additional activities for ASD improvement was 1314 USD, 1148 USD, and 1152 USD per year, with 21.1%, 22.4%, and 45.6% of patients engaging the activities for mild, moderate, and severe ASD patients, respectively.

Caregiver costs were also estimated, revealing that the average caregiver cost for accompanying patients for outpatient visits was 38 USD per visit, with annual visit averages 5.13, 4.89, and 6.65 for mild, moderate, and severe ASD patients, respectively. The caregiver cost for accompanying a patient during hospitalization was estimated at 225 USD per visit, with 34.6% of patients requiring hospitalization. The percentages of caregivers who needed to quit their job taking care of ASD patients were 17.8%, 13.5%, and 23.6% for mild, moderate, and severe ASD patients, respectively, with an average annual lost income of 518 USD. All cost data are detailed in Table 1.

Health utilities for different ASD severities were also estimated by the cross-sectional survey interview which were the same patients as in the direct nonmedical cost study. We found that the average utility was 0.87 ± 0.09 for mild ASD patients, while the average utilities for moderate and severe ASD patients were 0.84 ± 0.10 and 0.79 ± 0.14, respectively (Table 1).

### 2.6. Data Analysis

The total quality-adjusted life years (QALYs) and total lifetime costs were estimated. The annual discount rate of 3% was applied for both QALYs and costs. Incremental cost-effectiveness ratios (ICERs) were calculated to compare costs and QALYs associated with TDAS and ClinDx using the following formula:$$ICER=\frac{Total\;discounted\;cost\;of\;TDAS-Total\;discounted\;cost\;of\;ClinDx\;}{Total\;discounted\;QALY\;of\;TDAS-Total\;discounted\;QALY\;of\;ClinDx}$$

The base-case analysis was performed using the mean or point estimates of each input parameter to calculate ICER.

One-way sensitivity analysis was conducted to evaluate the effect of each input on the ICERs. Additionally, probabilistic sensitivity analysis involving 10,000 iterations was performed to explore the robustness of the main findings. The 95% CI of each input was used to represent the estimated uncertainty of the input when available. In cases where the 95% CI was not available, a variation of 20% was instead applied.

Pre-specified data distribution was employed. Beta or Dirichlet distributions were utilized for probability values within the decision tree, transitional probabilities, and utilities in the Markov model when appropriate. Gamma distribution was applied for cost data. The consumer price index was employed to convert past cost data to their current value.

Furthermore, a scenario analysis by changing the percentage of different severity in ClinDx was performed to explore the effect of such input on the findings. In this scenario analysis, the percentages of different severity in ClinDx were 11.39%, 45.57%, and 43.04% for mild, moderate, and severe ASD, respectively.

The willingness to pay (WTP) threshold was set at 160,000 Thai baht (THB)/QALY (4577 USD/QALY). This threshold was used as the benchmark to assess whether TDAS or ClinDx provide good value for money. The cost-effectiveness acceptability curve (CEAC) was generated by varying the WTP threshold. All costs expressed in THB were converted to USD using an exchange rate of 34.995 THB/QALY, according to a report for the Bank of Thailand on 18 December 2023. This conversion ensures that the economic evaluations are accessible and understandable in a global context.

## 3. Results

### 3.1. Base-Case and Scenario Analysis Findings

Our base-case analysis revealed that TDAS could yield an improvement of approximately 1.96 QALY compared to ClinDx. However, it was also associated with an increase in the total lifetime cost by 5577 USD, resulting in an ICER of 2852 USD/QALY.

In the scenario analysis where the percentage distribution of ASD severity in ClinDx was altered, the findings were consistent with the base-case findings. In the scenario analysis, TDAS could yield an improvement of 1.91 QALY compared to ClinDx, but at an increased total lifetime cost of 5907 USD. The ICER of the scenario analysis was 3092 USD/QALY (Table 2).

### 3.2. Sensitivity Analysis

The one-way sensitivity analysis showed that the probabilities of severe and moderate ASD diagnoses using TDA, as well as the probabilities of mild ASD diagnoses using both TDAS and ClinDx, were the top four influencing factors of the findings. This suggests that variations in these inputs significantly affect the ICERs.

Considering the uncertainty of these four key factors, the overall conclusion of our findings could change depending on their actual value. If these inputs were to shift in a certain way, the ICERS could transition from being less than the WTP threshold (indicating cost-effectiveness) to exceeding the WTP threshold (indicating non-cost-effectiveness). This change underscores the importance of these inputs and highlights the need for robustness of the inputs to ensure the reliability of the conclusion (Figure 2).

The probabilistic sensitivity analysis revealed that the majority of iterations (93.41%) fell into the right-upper quadrant, indicating that TDAS generally led to higher QALYs but also increased total lifetime costs compared to ClinDx. Additionally, 2.24% fell into the right lower quadrant, suggesting that TDAS was cost-saving. Conversely, only 2.14% of the iterations resulted lower QALYs but higher total lifetime costs for TDAS, placing the outcome in the less favorable left-upper quadrant. Crucially, TDAS was found to have an 81.16% possibility of being cost-effective at the current WTP threshold (Figure 3). This high percentage indicates strong evidence supporting the cost-effectiveness of TDAS compared to ClinDx under the current WTP.

## 4. Discussions

This study demonstrated the long-term clinical and economic benefits of using TDAS over ClinDx for patients suspected of developmental delays aged 1–5 years. TDAS has been shown to facilitate early diagnosis of ASD, resulting in an improvement in total life years and QALYs. Specifically, TDAS could provide a 3.92 undiscounted QALY gain or 1.96 discounted QALY gain, although it requires a higher lifetime cost of 5577 USD compared to ClinDx. This results in an ICER of 2852 USD/QALY. Given these findings, it can be concluded that TDAS represents a cost-effective option for ASD diagnosis in Thailand, when considering the current WTP threshold. This suggests that implementing TDAS could be a viable option for improving ASD management in children, offering both clinical and economic benefits.

The finding suggests that TDAS is cost-effective, likely due to its potential for early detection of ASD. Children diagnosed early through TDAS might exhibit less severe disease symptoms compared to those diagnosed through ClinDx. Consequently, this early detection leads to lower healthcare costs and higher health utility and QALY in patients diagnosed with TDAS compared to those diagnosed with ClinDx.

Although TDAS is a relatively new diagnostic tool, it has been proven to be an effective tool for ASD detection. A previous study [18] demonstrated that TDAS possesses good overall content validity, with item-objective congruence scores ranging from 0.71 to 1.00, and good construct validity, with Tucker-Lewis indexes of 0.882 for the observation section and 0.858 for the interview section. Furthermore, TDAS showed excellent sensitivity at 100% and a good specificity at 82.4%. Another study [17] indicated that the performance of TDAS is comparable to that of the ADOS-2, a well-established instrument in ASD diagnosis. These highlight the potential of TDAS as a reliable and effective option for early ASD diagnosis.

Our study used ClinDx as a comparator because it represents the standard clinical practice for ASD diagnosis in Thailand, despite the availability of other diagnostic tools such as ADOS-2 [10]. ADOS-2 has a limitation in terms of the accessibility and potential cultural discrepancies that may affect its applicability and interpretation within Thai context. Consequently, the choice to use ClinDx as the comparator is acceptable and practical for the scope of study, ensuring that the findings are relevant and directly applicable to the current clinical practice in Thailand.

Most inputs used to inform the model were from the study-specific data collection, including a cross-sectional survey of patients or caregivers to estimate direct nonmedical costs and utility, a cross-section interview of healthcare providers to estimate time spent for TDAS and ClinDx leading to the estimated labor cost, and a retrospective database analysis to estimate direct medical cost. This approach emphasizes the use of real-world data to inform the model, which is instrumental in reflecting actual clinical practice and conditions more accurately than models solely based on a literature review.

The study calculated the cost of TDAS based on experiences gathered from its use in a pilot project. The cost of TDAS implementation consisted of TDAS training, TDAS instruments, and the labor cost of healthcare providers for utilizing TDAS. The cost of TDAS training might vary significantly with a nationwide implementation. Specifically, the cost might decrease due to the potential organization of multiple training sessions throughout the country, which could reduce travel and associated expenses for both trainers and trainees. Additionally, with nationwide adoption, the increased demand for TDAS instruments could lead to lower costs per unit of the instruments. Such reductions in the cost of training and instruments, as a result of broader implementation, would contribute to lowering the overall cost of TDAS implementation.

Our sensitivity analyses underscore the importance of ASD severities, which were diagnosed by both TDAS and ClinDx in the model. The cost-effectiveness status of TDAS could be different if the probabilities of being classified as mild, moderate, and severe ASD were changed. In this study, the probabilities of different severities of ASD by both TDAS and ClinDx were derived from our retrospective medical records review. The hospitals were selected because they have been participating in the pilot project of TDAS implementation. Among these, one is a specialty hospital focused on child development, which could introduce a referral bias, as patients with more severe conditions are likely more frequent at this specialized institution compared to general hospitals. To mitigate potential bias, we re-analyzed the proportion of severity excluding data from the specialty hospital. This adjustment led to a decrease in the proportion of being classified as severe ASD by ClinDx from 60.81% to 43.04%. Despite the changes in total lifetime costs and QALY improvements for ClinDx, the ICER remained similar to the main findings, demonstrating the robustness of our findings.

Our probabilistic sensitivity analysis indicated that there is more than an 80% possibility of TDAS being cost-effective at the current WTP. According to our CEAC, TDAS could be more cost-effective than ClinDx at the WTP at approximately 3000 USD/QALY. This suggests that even though the WTP is 1.5 times lower than the current WTP, TDAS still maintains a high likelihood of being cost-effective.

The limitations of this study need to be addressed. First, this study was based on our primary data collection from patients at five hospitals. The inputs might differ were the data obtained from other hospital settings. However, because TDAS is a new diagnostic tool, only clinicians and healthcare providers from the five hospitals have experience with TDAS; hence, collecting data from other hospitals for TDAS would not be feasible. Although data for ClinDx could be collected from hospitals other than the included hospitals, patients’ characteristics and hospital policies and guidelines for ASD might differ from the included hospital. Therefore, data for ClinDx from the same settings could be more suitable to inform the model. Thus, further studies are warranted to estimate cost and utility of ASD patients using TDAS and ClinDx in the same settings and contexts. Second, the labor costs used to inform the model were derived from interviews of healthcare providers about ASD diagnosis and time spent performing TDAS. The labor costs might vary from one setting to another; therefore, the costs of TDAS and ClinDx might differ across different settings. However, according to our sensitivity analysis, since labor cost is not a significant influencing factor, variations in labor costs might not significantly impact the cost-effectiveness findings. Lastly, because there are many interventions and treatments for children with ASD, this model did not consider different interventions and treatments in the follow-up phase. Instead, we applied the regression and progression of ASD severity using real-world data from the studied hospitals. The regression and progression of ASD severity could vary according to different interventions and treatments among hospitals. Therefore, follow-up monitoring and evaluation are necessary when TDAS is implemented.

The implementation of TDAS could benefit not only physicians by providing a diagnostic tool for ASD but also patients suspected of development delays in general. TDAS requires trained healthcare providers to perform the evaluation, but the healthcare providers could be nurses, occupational therapists, psychologists, or speech therapists. It does not necessitate the involvement of child and adolescent psychiatrists or child developmental pediatricians to perform the evaluation, professionals who are limited in number. Therefore, TDAS could likely improve early ASD detection as it could foster the ASD detection process and increase patient accessibility to ASD assessments. However, implementing TDAS requires additional health resources and management, such as TDAS training, additional healthcare providers for TDAS evaluation, and other infrastructures. Thus, national policy decision-makers might need to consider TDAS as an option for early ASD detection, accompanied by a comprehensive plan for effective implementation.

## 5. Conclusions

TDAS is the cost-effective option for ASD diagnosis in Thai children aged 1–5 years compared to ClinDx, despite some uncertainties around inputs. Further monitoring and evaluation are warranted if TDAS is to be implemented nationwide.

## Figures and Tables

**Figure 1 healthcare-12-00782-f001:**
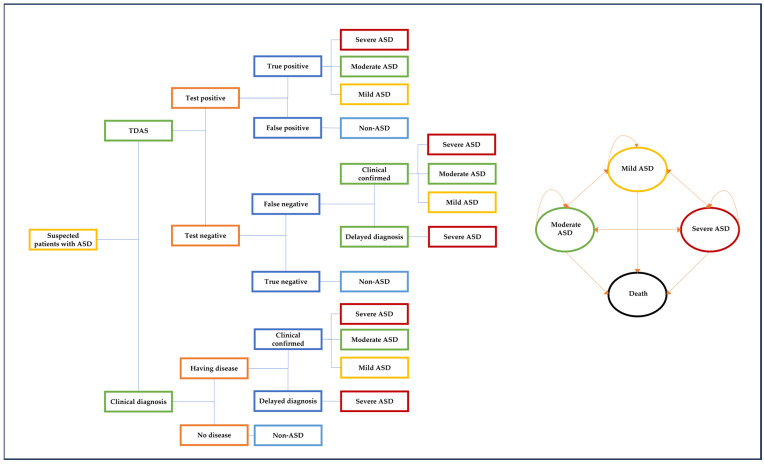
A hybrid model consisting of a decision tree and a Markov model.

**Figure 2 healthcare-12-00782-f002:**
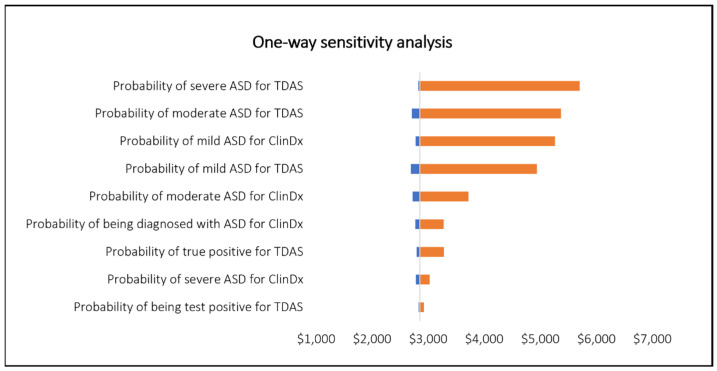
One-way sensitivity analysis.

**Figure 3 healthcare-12-00782-f003:**
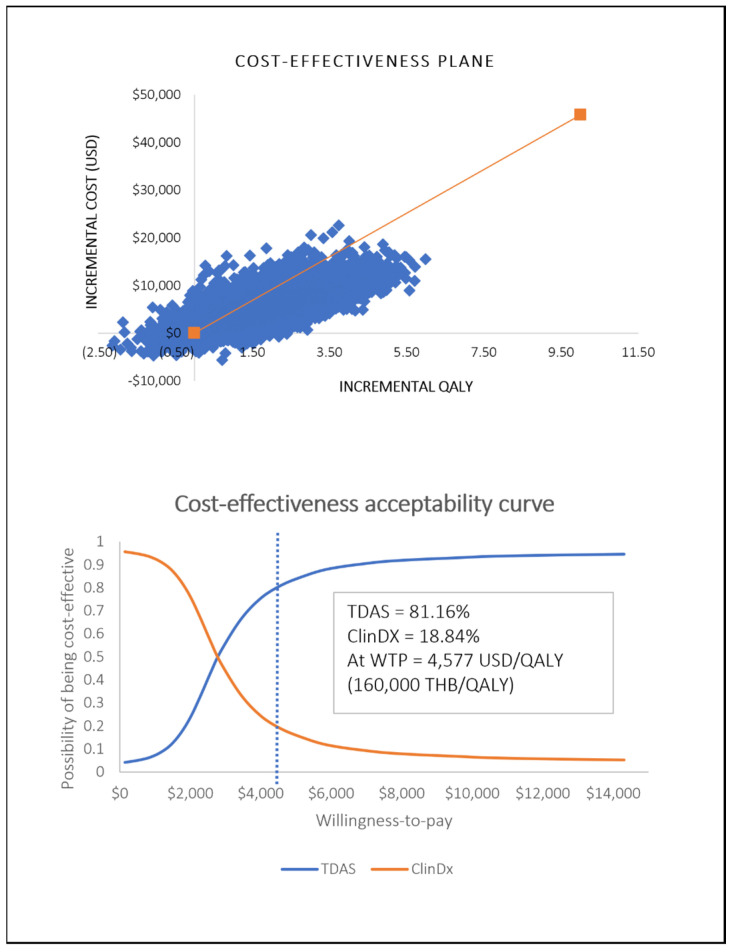
Probabilistic sensitivity analysis and cost-effectiveness acceptability curve.

**Table 1 healthcare-12-00782-t001:** Input parameters.

Input Parameters	Value (Standard Error)	Distribution	References
**Probabilities for diagnostic phase (Decision tree model)**			
Probability of being test-positive for TDAS	0.8990 (0.0339)	Beta	[27]
Probability of true positive for TDAS	0.8286 (0.0368)	Beta	[17]
Probability of false positive for TDAS	0.1961 (0.0556)	Beta	[17]
Probability of delayed diagnosis for TDAS	0.0071 (0.0071)	Beta	A retrospective chart review from 5 hospitals
Probability of mild ASD for TDAS	0.1111 (0.0786)	Dirichlet	
Probability of moderate ASD for TDAS	0.5069 (0.0585)	Dirichlet	
Probability of severe ASD for TDAS	0.3819 (0.0655)	Dirichlet	
Probability of being diagnosed with ASD for ClinDx	0.6731 (0.0376)	Beta	[17]
Probability of delayed diagnosis for ClinDx	0.2000 (0.0332)	Beta	A retrospective chart review from 5 hospitals
Probability of mild ASD for ClinDx	0.0608 (0.0797)	Dirichlet	
Probability of moderate ASD for ClinDx	0.3311 (0.0672)	Dirichlet	
Probability of severe ASD for ClinDx	0.6081 (0.0515)	Dirichlet	
**Probabilities for follow-up phase (Markov model)**			
Transitional probabilities from mild to moderate	0.0620 (0.0538)	Beta	A retrospective chart review from 5 hospitals
Transitional probabilities from mild to severe	0.0000 (0.0000)	Fixed	
Transitional probabilities from moderate to mild	0.2643 (0.0226)	Beta	
Transitional probabilities from moderate to severe	0.0218 (0.0087)	Beta	
Transitional probabilities from severe to mild	0.0025 (0.0044)	Beta	
Transitional probabilities from severe to moderate	0.2078 (0.1730)	Beta	
Relative risk of death in patients with ASD	2.370 (0.0942)	Log-normal	[28]
**Utility values**			
Utility for mild ASD	0.8659 (0.0085)	Beta	A cross-sectional interview by EQ-5D-Y from 5 hospitals
Utility for moderate ASD	0.8410 (0.0098)	Beta	
Utility for severe ASD	0.7930 (0.0177)	Beta	
**Direct medical cost (USD)**			
Cost of ASD treatment for an inpatient visit (mild)	387 (88)	Gamma	A retrospective database analysis from 5 hospitals
Cost of ASD treatment for an inpatient visit (moderate)	416 (38)	Gamma	
Cost of ASD treatment for an inpatient visit (severe)	664 (53)	Gamma	
Proportion of patients having admission	0.3460 (0.0280)	Beta	A cross-sectional interview from 5 hospitals
Proportion of patient hospitalized with mild ASD	0.0585 (-)	Fixed	
Proportion of patient hospitalized with moderate ASD	0.3032 (-)	Fixed	
Proportion of patient hospitalized with severe ASD	0.6383 (-)	Fixed	
Cost of outpatient ASD treatment (mild)	66 (11)	Gamma	A retrospective database analysis from 5 hospitals
Cost of outpatient ASD treatment (moderate)	133 (7)	Gamma	
Cost of outpatient ASD treatment (severe)	124 (12)	Gamma	
Cost of purchasing additional instruments (mild)	11 (26)	Gamma	A cross-sectional interview from 5 hospitals
Cost of purchasing additional instruments (moderate)	197 (24)	Gamma	
Cost of purchasing additional instruments (severe)	199 (25)	Gamma	
Proportion of patients purchasing additional instruments (mild)	0.5565 (-)	Fixed	
Proportion of patients purchasing additional instruments (moderate)	0.6852 (-)	Fixed	
Proportion of patients purchasing additional instruments (severe)	0.8966 (-)	Fixed	
Cost of additional activities for ASD (mild)	1314 (246)	Gamma	
Cost of additional activities for ASD (moderate)	1148 (188)	Gamma	
Cost of additional activities for ASD (severe)	1152 (216)	Gamma	
Proportion of attending additional activities (mild)	0.2114 (-)	Fixed	
Proportion of attending additional activities (moderate)	0.2243 (-)	Fixed	
Proportion of attending additional activities (severe)	0.4561 (-)	Fixed	
**Direct non-medical cost (USD)**			
Travel cost per outpatient visit	13 (0.9)	Gamma	A cross-sectional interview from 5 hospitals
Additional food cost per outpatient visit	8 (0.5)	Gamma	
Accommodation cost per outpatient visit	24 (3.6)	Gamma	
Proportion of patients with accommodation for outpatient	0.60 (0.03)	Beta	
Average number of outpatient visits per year (mild)	5.13 (0.49)	Gamma	
Average number of outpatient visits per year (moderate)	4.89 (0.48)	Gamma	
Average number of outpatient visits per year (severe)	6.65 (0.91)	Gamma	
Caregiver cost for accompanying patients to outpatient visit	38 (3)	Gamma	
Travel cost per admission for patients	48 (6)	Gamma	
Additional food cost per admission for patients	9 (0.7)	Gamma	
Accommodation cost per admission for patients	15 (1)	Gamma	
Proportion of patients with accommodation for admissions	0.385 (0.078)	Beta	
Travel cost per admission for caregivers	63 (10)	Gamma	
Additional food cost per admission for caregivers	80 (8)	Gamma	
Accommodation cost per admission for caregivers	183 (48)	Gamma	
Caregiver cost for accompanying patients to admissions	225 (105)	Gamma	
Average number of admissions per year	1 (-)	Fixed	
**Caregiver cost at home (USD)**			
Hired caregiver salary	196 (32)	Gamma	A cross-sectional interview from 5 hospitals
Proportion of hired caregiver	0.058 (-)	Fixed	
Unhired caregiver salary	518 (53)	Gamma	
Proportion of unhired caregiver quitting a job (mild)	0.1778 (-)	Fixed	
Proportion of unhired caregiver quitting a job (moderate)	0.1348 (-)	Fixed	
Proportion of unhired caregiver quitting a job (severe)	0.2364 (-)	Fixed	
**Cost of TDAS and ClinDx**			
Cost of TDAS training (per provider)	151 (±20%)	Gamma	Internal data from previous TDAS implementation
Cost of TDAS instruments at implementation	323 (±20%)	Gamma	
Cost of TDAS maintenance per patient per year	0.27 (±20%)	Gamma	
Cost of cloud services per patient per year	1.43 (±20%)	Gamma	
Average labor cost for TDAS (per month)	134 (±20%)	Gamma	An interview from healthcare providers
Average labor cost for ClinDx (per month)	61 (±20%)	Gamma	
The number of ASD diagnoses (per week)	2.0 (-)	Fixed	

**Table 2 healthcare-12-00782-t002:** Base-case analysis and Scenario analysis findings.

Intervention	Undiscounted					Discounted				
	Cost (USD)	QALY	Incremental Cost (USD)	Incremental QALY	ICER (USD/QALY)	Cost (USD)	QALY	Incremental Cost (USD)	Incremental QALY	ICER (USD/QALY)
**Base-case analysis**										
**TDAS**	80,304	31.36	10,119	3.92	2579	43,231	15.55	5577	1.96	2852
**ClinDx**	70,185	27.43	Reference			37,654	13.59	Reference		
**Scenario analysis**										
**TDAS**	80,293	31.36	10,408	3.84	2709	43,219	15.55	5907	1.91	3092
**ClinDx**	69,885	27.52	Reference			37,312	13.64	Reference		

## Data Availability

Data available upon appropriate request to corresponding author due to privacy and ethical restrictions.

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
