# Peer review of "Economic Evaluation of the Thai Diagnostic Autism Scale for Autism Spectrum Disorder Diagnosis in Children Aged 1–5 Years Old"

_healthcare, 2024, doi:10.3390/healthcare12070782_

Round 1

Reviewer 1 Report

Comments and Suggestions for Authors

This study assessed the cost-effectiveness of TDAS compared to clinical diagnosis (ClinDx) for ASD diagnosis in Thai children aged 1 – 5 years from a societal perspective with a hybrid model consisted of a decision tree model for a diagnostic phase and a state transition model for a follow-up phase. It found that TDAS could improve QALY by 1.96 but it increased the total lifetime cost by 3,307 USD, resulting in the ICER as 1,691 USD/QALY. This research result has good application value.

However, since the authors did not introduce the existing studies on the cost-effectiveness analysis of autism measurement tools in the introduction, it is impossible to judge the innovation of this study. Therefore, it is suggested that the authors add this part.

The results showed that TDAS was the cost-effective option for ASD diagnosis in Thai children aged 1 – 5 years compared to ClinDx, but the authors did not explore the possible causes, which need to be added to the discussion.

In addition, since autistic children diagnosed with TDAS and autistic children diagnosed with ClinDx are two different groups, the cost-effectiveness of comparing two different groups may be affected by the economic strength and intervention willingness of these two groups. Therefore, It is suggested that in future studies, the same group of children with autism be diagnosed with both TDAS and ClinDx, and then compare the cost-effectiveness of the two diagnostic tools.

Author Response

Response to reviewer 1 comments

Comment 1

This study assessed the cost-effectiveness of TDAS compared to clinical diagnosis (ClinDx) for ASD diagnosis in Thai children aged 1 – 5 years from a societal perspective with a hybrid model consisted of a decision tree model for a diagnostic phase and a state transition model for a follow-up phase. It found that TDAS could improve QALY by 1.96 but it increased the total lifetime cost by 3,307 USD, resulting in the ICER as 1,691 USD/QALY. This research result has good application value.

However, since the authors did not introduce the existing studies on the cost-effectiveness analysis of autism measurement tools in the introduction, it is impossible to judge the innovation of this study. Therefore, it is suggested that the authors add this part.

Response 1

                We agree that adding information on the cost-effectiveness of other ASD diagnosis tools might improve the comprehensiveness of information for ASD diagnosis. We added some information to the previous CEA, as shown below.

                Original version

                (none)

                Revised version

Previous cost-effectiveness studies have shown that early detection and providing interventions before definite ASD diagnosis could enhance long-term clinical outcomes and be cost-effective, especially for high-risk population.        

Comment 2

The results showed that TDAS was the cost-effective option for ASD diagnosis in Thai children aged 1 – 5 years compared to ClinDx, but the authors did not explore the possible causes, which need to be added to the discussion.

Response 2

We added a discussion to describe the possible causes of the finding as shown below.

                Original version

                (none)

Revised version

The finding suggests that TDAS is cost-effective, likely due to its potential for early detection of ASD. Children diagnosed early through TDAS might exhibit less severe disease symptoms compared to those diagnosed through ClinDx. Consequently, this early detection leads to lower healthcare costs and higher health utility and QALY in patients diagnosed with TDAS compared to those diagnosed with ClinDx.

Comment 3

In addition, since autistic children diagnosed with TDAS and autistic children diagnosed with ClinDx are two different groups, the cost-effectiveness of comparing two different groups may be affected by the economic strength and intervention willingness of these two groups. Therefore, it is suggested that in future studies, the same group of children with autism be diagnosed with both TDAS and ClinDx, and then compare the cost-effectiveness of the two diagnostic tools.

Response 3

We revised our limitations to address this concern as shown below.

                Original version

Limitations of this study should be addressed. First, this study was based on our primary data collection from patients at five hospitals. The inputs might be different if data was obtained from other hospital settings. However, because TDAS is a newly diagnostic tool, only clinicians and healthcare providers from the five hospitals have experienced TDAS, it would not be possible to collect data from other hospitals for TDAS. Although data for ClinDx could be collected from other hospitals rather than the five included hospitals, patients’ characteristics and hospital policies and guidelines for ASD might be different from the included hospital. Therefore, data for ClinDx from the same settings could be more suitable to inform the model. Second, the labor cost used to inform the model was from an interview of healthcare providers for ASD diagnosis and time spent for performing TDAS. The labor cost might vary from one to one, therefore, cost of TDAS and ClinDx might be different from settings to settings. However, according to our sensitivity analysis, labor cost is not a significant influencing factor, the variation in labor cost might not have a significant effect on cost-effectiveness findings. Last, because there have been many interventions and treatments for children with ASD, this model did not consider different interventions and treatments in the follow-up phase of the model. Instead, we applied the regression and progression of ASD severity using real-world data from the studied hospitals. The regression and progression of ASD severity could vary according to different interventions and treatments among hospitals. Therefore, the follow-up monitoring and evaluation is needed when TDAS is implemented.

                Revised version

The limitations of this study need to be addressed. First, this study was based on our primary data collection from patients at five hospitals. The inputs might differ if data was obtained from other hospital settings. However, because TDAS is a new diagnostic tool, only clinicians and healthcare providers from the five hospitals have experience with TDAS; hence, collecting data from other hospitals for TDAS would not be feasible. Although data for ClinDx could be collected from hospitals other than the included hospitals, patients’ characteristics and hospital policies and guidelines for ASD might differ from the included hospital. Therefore, data for ClinDx from the same settings could be more suitable to inform the model. Thus, further studies are warranted to estimate cost and utility of ASD patients using TDAS and ClinDx in the same settings and contexts. Second, the labor costs used to inform the model was derived from interviews of healthcare providers about ASD diagnosis and time spent performing TDAS. The labor costs might vary from one setting to another; therefore, the costs of TDAS and ClinDx might differ across different settings. However, according to our sensitivity analysis, since labor cost is not a significant influencing factor, variations in labor costs might not significantly impact the cost-effectiveness findings. Lastly, because there are many interventions and treatments for children with ASD, this model did not consider different interventions and treatments in the follow-up phase. Instead, we applied the regression and progression of ASD severity using real-world data from the studied hospitals. The regression and progression of ASD severity could vary according to different interventions and treatments among hospitals. Therefore, follow-up monitoring and evaluation are necessary when TDAS is implemented

Reviewer 2 Report

Comments and Suggestions for Authors

The major flaw of this study is the lack of statements on the reasons for conducting this study. It is a matter of course that performing TDAS requires more resources, such as additional healthcare providers, instruments during observation, and cloud storage for recording video during observation, resulting in an additional healthcare cost for ASD detection. It is the price of obtaining a reliable diagnosis of ASD. The clinics with manpower should perform such kind of evaluation for any child with a suspected ASD. It is unnecessary to conduct a study to determine the cost-effectiveness of the implementation of TDAS. Given that this study compared the cost-effectiveness between TDAS and only clinical diagnosis, the advantages of TDAS in outcomes and disadvantages of TDAS in costs can be expected. I would like to see the comparisons of advantages and disadvantages between TDAS and ADOS-2.

Comments on the Quality of English Language

there are a lot of typos.

Author Response

Response to reviewer 2 comments

Comment 1

The major flaw of this study is the lack of statements on the reasons for conducting this study. It is a matter of course that performing TDAS requires more resources, such as additional healthcare providers, instruments during observation, and cloud storage for recording video during observation, resulting in an additional healthcare cost for ASD detection. It is the price of obtaining a reliable diagnosis of ASD. The clinics with manpower should perform such kind of evaluation for any child with a suspected ASD. It is unnecessary to conduct a study to determine the cost-effectiveness of the implementation of TDAS. Given that this study compared the cost-effectiveness between TDAS and only clinical diagnosis, the advantages of TDAS in outcomes and disadvantages of TDAS in costs can be expected. I would like to see the comparisons of advantages and disadvantages between TDAS and ADOS-2.

Response 1

We understand that a comparison between TDAS and ADOS could benefit the policy decision in some contexts. However, based on Thai context, we used ClinDx as the standard clinical practice. Our policy makers are considering adding TDAS into healthcare benefit packages compared to ClinDx rather than ADOS-2. Moreover, even though TDAS provide better clinical outcomes with higher cost as expected, it is still a need to evaluate whether the higher cost of TDAS is worth investing for better clinical outcomes compared to our societal willingness to pay (WTP).

We added the rationale in the introduction as shown below. In addition, we originally discussed the selection of ClinDx as a comparator as shown below.

Original version

In Thailand, ADOS-2 is widely used as a diagnostic tool to identify children with ASD. However, there is an argument that the translated version of the Western diagnostic tools might not be valid due to the differences in cultural factors. Thus, a new ASD diagnostic tool called the Thai Diagnostic Autism Scale (TDAS) has been developed and validated for assessing suspected children with ASD.[13,14] TDAS consists of 13 times for observation and 17 items for interview. The studies demonstrate that TDAS is a valid tool to diagnose children with ASD. The sensitivity and specificity of TDAS are 82.86% and 80.93%, respectively. The accuracy of TDAS is estimated at 82.05%. However, TDAS requires more resources, such as additional healthcare providers, instruments during observation, and cloud storage for recording video during observation, resulting in an additional healthcare cost for ASD detection. The cost-effectiveness of the implementation of TDAS should be evaluated. Therefore, this study aimed to assess the cost-effectiveness of TDAS compared to clinical diagnosis (ClinDx) for ASD diagnosis for Thai children aged 1 – 5 years from a societal perspective.

Revised version

In Thailand, the ADOS-2 is one of the ASD diagnosis tools to identify children with ASD. However, there are concerns regarding the validity of translated version of the Western diagnostic tools due to cultural differences and it is limited use only in some settings. According to the limitations of ADOS-2, clinical diagnosis (ClinDx) based on the fifth edition of the Diagnostic and Statistical Manual of Mental Disorders (DSM-5) remains the standard practice for ASD diagnosis in the country.

A novel ASD diagnostic tool, the Thai Diagnostic Autism Scale (TDAS), has been developed and validated for assessing children suspected of having ASD. TDAS comprises 13 observation items and 17 interview items. TDAS is a valid tool to diagnose children with ASD with a sensitivity of 82.86% and a specificity of 80.93%. Despite its accuracy of 82.05%, TDAS requires additional resources, such as healthcare providers, instruments during observation, and cloud storage for recording video during observation, resulting in an additional healthcare cost for ASD detection. Thus, the cost-effectiveness of the implementation of TDAS should be evaluated. At the time of conducting this study, the National Health Security Office (NHSO), a public healthcare payer in Thai healthcare system covering approximately 70% of Thai citizens, is considering including TDAS into its health benefit packages for Thai citizens under the Universal Coverage Schemes (UCS). The NHSO requires economic evidence, especially cost-effectiveness analysis to evaluate the value for money of TDAS for ASD diagnosis. This study aimed to assess the cost-effectiveness of TDAS compared to clinical diagnosis (ClinDx) for ASD diagnosis for Thai children aged 1 – 5 years from a societal perspective.

                Original version

Our study used ClinDx as a comparator of this study because ClinDx is the standard clinical practice to diagnose ASD in Thailand, even though some ASD diagnostic tools are available such as ADOS-2. ADOS-2 has a limitation in terms of the accessibility of the tool and differences in cultural effects. Thus, we believe the use of ClinDx as the comparator is acceptable and practical in Thailand.

Revised version

Our study used ClinDx as a comparator because it represents the standard clinical practice for ASD diagnosis in Thailand, despite the availability of other diagnostic tool such as ADOS-2. ADOS-2 has a limitation in terms of the accessibility and potential cultural discrepancies that may affect its applicability and interpretation within Thai context. Consequently, the choice to use ClinDx as the comparator is acceptable and practical for the scope of study, ensuring that the findings are relevant and directly applicable to the current clinical practice in Thailand.

Round 2

Reviewer 1 Report

Comments and Suggestions for Authors

The authors added information on the cost-effectiveness of other ASD diagnosis tools. However, the innovation of this study needs to be clearly mentioned. For example, did existing studies examine the cost-effectiveness of TDAS? is there any new method used for cost-effectiveness analysis in this study?

Figure 1 is fuzzy and needs to be drawn clearly or enlarged.

Author Response

Comment#1

The authors added information on the cost-effectiveness of other ASD diagnosis tools. However, the innovation of this study needs to be clearly mentioned. For example, did existing studies examine the cost-effectiveness of TDAS? is there any new method used for cost-effectiveness analysis in this study?

Response#1

We add more information on the CEA of TDAS in the introduction.

            Revision 1

A novel ASD diagnostic tool, the Thai Diagnostic Autism Scale (TDAS), has been developed and validated for assessing children suspected of having ASD.[17,18] TDAS comprises 13 observation items and 17 interview items. TDAS is a valid tool to diagnose children with ASD with a sensitivity of 82.86% and a specificity of 80.93%. Despite its accuracy of 82.05%, TDAS requires additional resources, such as healthcare providers, instruments during observation, and cloud storage for recording video during observation, resulting in an additional healthcare cost for ASD detection. Thus, the cost-effectiveness of the implementation of TDAS should be evaluated.

Revision 2

A novel ASD diagnostic tool, the Thai Diagnostic Autism Scale (TDAS), has been developed and validated for assessing children suspected of having ASD.[17,18] TDAS comprises 13 observation items and 17 interview items. TDAS is a valid tool to diagnose children with ASD, with a sensitivity of 82.86% and a specificity of 80.93%. Despite its accuracy of 82.05%, TDAS requires additional resources, such as healthcare providers, instruments during observation, and cloud storage for recording video during observation, resulting in an additional healthcare cost for ASD detection. To date, information on the cost-effectiveness of TDAS for ASD diagnosis is not available. Cost-effectiveness information is important information for national healthcare policy decision-making.

Revision 1

This study aimed to assess the cost-effectiveness of TDAS compared to clinical diagnosis (ClinDx) for ASD diagnosis for Thai children aged 1 – 5 years from a societal perspective.

Revision 2

This study aimed to assess the cost-effectiveness of TDAS compared to ClinDx for ASD diagnosis for Thai children aged 1 – 5 years from a societal perspective using a standard decision tree with a Markov model.

Comment#2

Figure 1 is fuzzy and needs to be drawn clearly or enlarged.

Response#2

We re-draw Figure 1 for more clarity.

Reviewer 2 Report

Comments and Suggestions for Authors

The authors have revised their manuscript based on the reviewer's suggestions. I would like to suggest the editors accepting it for publication.

Author Response

We appreciate your support.